# Therapeutic effects of skin repair plus ointment on molecular and morphological changes in burn wounds: A rat model

Foad Alzoughool[1,2]*, Mohammad Borhan Al-Zghoul[3], Wael Hananeh[4], Manar Atoum[1], Reman Alnemrat[1], Mohammad Alzghool[5], Ahmad M. Al-Bashaireh[6], Yousef Aljawarneh[2], Mohamad Mayyas[7]

1 Department of Medical Laboratory Sciences, Faculty of Applied Medical Sciences, The Hashemite University, Zarqa, Jordan, 2 Department of Nursing, Faculty of Health Sciences, Higher Colleges of Technology, Abu Dhabi, United Arab Emirates, 3 Basic Veterinary Sciences, School of Veterinary Medicine, Jordan University of Science and Technology, Irbid, Jordan, 4 Veterinary Pathology and Public Health, School of Veterinary Medicine, Jordan University of Science and Technology, Irbid, Jordan, 5 Faculty of Medicine, Wuhan University, Wuhan, China, 6 Faculty of Nursing, Philadelphia University, Amman, Jordan, 7 Department of Animal Production, Faculty of Agriculture, Jordan University of Science and Technology, Irbid, Jordan

* foad@hu.edu.jo

## Abstract

Skin Repair Plus Ointment (Remanco) is an advanced formulation of the traditional Jordanian cream, which has been used for treating burn injuries and has served as an alternative remedy in northern Jordan for several decades. In this study, we sought to investigate the therapeutic effects of Remanco on the healing morphology and molecular alterations associated with burn wounds. This randomized controlled experiment involved 90 Wistar rats (160−170 grams) that were randomly assigned to three groups of 30 rats each. The groups were designated as follows: negative control (burn treated with normal saline), positive control (burn treated with 1% silver sulfadiazine), and tested (burn treated with Remanco cream). Burn injuries were inflicted on the shaved dorsum of the rats using a preheated 30-mm-wide circular copper head device. The burns were treated daily with normal saline, 1% silver sulfadiazine, or Remanco cream for 30 days or until the animals were sacrificed. Gross evaluation, histopathological assessment, and RT-PCR Analysis were conducted to evaluate the efficacy of Remanco cream in promoting wound healing. Although there was no statistically significant difference in wound area measurements or histopathological ratings, the cosmetic appearance of the wound area exhibited a significant improvement in the Remanco cream-treated group. Molecular investigations revealed that the Remanco-treated group demonstrated a significant increase in the levels of two essential growth factors: TGF-beta on day 3 and IGF-1 on day 9, with a p-value less than 0.05. In summary, Remanco cream holds potential for enhancing wound healing in animal models, but further studies are necessary to investigate its potential clinical applications.

**Data availability statement:** The datasets supporting this study have been deposited in Figshare and are publicly available at 10.6084/m9.figshare.30598193.

**Funding:** Funding was obtained from Hashemite University/Deanship of research, fund No. 29/2020. The funders had no role in study design, data collection and analysis, decision to publish, or preparation of the manuscript.

**Competing interests:** The authors have declared that no competing interests exist.

## Introduction

Burn injury is defined as a traumatic injury primarily of thermal origin that affects tissues. Burn injuries are classified based on the depth of the affected tissue; hence, they can be superficial, partial-thickness, or full-thickness burns. First-degree burns affect only the epidermis, while second-degree burns are classified when the lesions penetrate the dermis and result in partial-thickness burns [1]. When burns have deep partial thickness, they can cause significant dermal damage that heals slowly and is associated with scar formation and potential functional loss [2].

Remanco is a modified version of the traditional Jordanian cream used for treating burns, commonly referred to as the "red cream" in northern Jordan. This cream has been utilized for decades as an alternative medicine to treat various skin problems, including burns, wounds, acne, diabetic ulcers, itching, and surgical scars in the countryside north of Jordan, specifically in Irbid. The main components of this cream are sesame oil, beeswax, myrrh, and Alkanna tinctoria. The literature indicates that each ingredient has various beneficial effects, such as anti-inflammatory properties, an accelerated wound healing process, and antimicrobial activity. However, no studies have examined the impact of all these components together.

The herbal ointment, which contains sesame oil, honey, and camphor, has been shown to effectively promote healing in rats with second-degree burn wounds [3,4]. It was also proposed that a diet rich in sesame oil could serve as an effective non-pharmacological treatment for atherosclerosis by managing inflammation and regulating lipid metabolism, shedding light on sesame oil's anti-inflammatory effects on the inflammation process [5].

The literature also indicates that the application of beeswax, olive oil, and Alkanna tinctoria to second-degree burns accelerates the epithelialization process, reduces pain during dressing changes, and leads to shorter hospital stays for patients [6]. It was also demonstrated that beeswax mixed with olive oil and butter could enhance burn wound healing and promote skin renewal by modulating tissue TGF-b1 and VEGF-a in a rat model of second-degree burns [7].

Myrrh, a natural gum or resin derived from small, thorny tree species of the genus Commiphora, has been used extensively in traditional medicine since approximately 2800 BCE for treating wounds, skin sores, inflammation, urinary tract diseases, and dental issues [8–10]. The study indicated that Myrrh essential oil demonstrates significant potential as a natural antimicrobial agent [9]. The effects of myrrh extract on the biological characteristics of human dermal fibroblasts (HDFs) have been investigated to explore its potential mechanisms in promoting wound healing. Myrrh water extract can significantly elevate the proliferation of Fb, accelerate the cell cycle of Fb, and upregulate the mRNA expression of type III collagen in Fb, which may be related to its mechanisms in promoting wound healing [11]. A study investigated the role of Myrrh and other herbal extracts on excision wound models, observing various physical, histological, and antimicrobial activities. It suggested that the healing effects resulted from stimulating faster collagen deposition, forming other connective tissue constituents, and exhibiting antibacterial activity [12]. Interestingly, Myrrh inhibits itch-associated histamine in stimulated mast cells, providing new evidence of its potential use for treating itch [13].

Numerous studies have indicated that Alkanna tinctoria, an herb classified within the borage family, possesses roots that serve as a dye. This herb has garnered recognition for its anti-aging properties, which notably impede in vitro lipid peroxidation of hepatic microsomal membranes, alongside its potential to inhibit or eradicate multidrug-resistant bacteria [14–16].

As previously mentioned, the individual components of this traditional cream have demonstrated numerous beneficial effects, including anti-inflammatory properties, enhanced wound healing, and antimicrobial activity. Nevertheless, no studies have examined the collective impact of all components. This study aims to investigate the role of a traditional cream containing these components in a rat model of second-degree burn injury.

## Methods

### Study design and subjects

This randomized controlled trial was conducted in accordance with the principles and regulations of laboratory animal care at Hashemite University. Approval from the institutional review board (IRB) was obtained from Hashemite University (Approval #: 2020/2019-5/2020). Remanco was approved as a cosmetic by the Jordan Food and Drug Administration (JFDA) (approval #: 14700/1/19/2).

### Experimental population

A total of ninety Wistar-Albino rats, with a weight range of 160–170 grams. Rats were obtained from the Experimental Animal House, Faculty of Veterinary Medicine, Jordan University of Science and Technology. Rats were housed in individual cages with ad libitum access to feed and water in an air-conditioned room (22–24ºC, 55–56% humidity) with a 12-h light and 12-h dark cycle. Rats were randomly allocated to three experimental groups, with thirty animals allocated to each group; each rat was housed in an individual cage.

**Group one: Negative control group.** Animals were subjected to burn injuries, and the burned lesions were treated with normal saline.

**Group two: Positive control group.** Animals were subjected to burns, and the burned lesions were treated with 1% silver sulfadiazine..

**Group three: Tested group.** Animals were subjected to burn, and the resulting lesions were treated with Remanco.

### Burns procedure and sample collection

Prior to the experiment, rats were housed in laboratory conditions for one week. All rats were maintained at a constant temperature of 22°C ± 2°C and subjected to a regulated 12-hour light/dark cycle. They were provided with conventional rat chow and had unrestricted access to water. Subsequently, the animals were weighed and administered intraperitoneal injections of ketamine (75 mg/kg) and xylazine (15 mg/kg) [17]. Electric clippers were employed to shave the backs of the anesthetized animals. Burn injuries to the backs were created using a laboratory-designed heating device, equipped with a 30-mm-diameter circular copper head. The heating device was heated in a boiling water bath for a duration of 10 minutes. Placing the heated device on the shaved back for a period of 20 seconds resulted in the desired burn wound [17].

### Treatment of wounds

In the treatment of burns after induction, the wounds were initially washed with sterile saline solution and subsequently dried using sterile gauze. Each group of rats received either normal saline (group 1, negative control group), 1% silver sulfadiazine (group 2, positive control group), or Remanco skin repair plus ointment (group 3, treated group). The treatments were administered evenly over the wound once daily for a period of 30 consecutive days or until the day of sacrifice. All groups underwent the same procedures. Throughout the experimental period, the wounds remained uncovered.

 

## Gross evaluation

A digital camera captured photographs of each animal's scorched area on days 1, 3, 6, 9, 15, 21, and 30 following the burn. A ruler was positioned on the wound's side to serve as a known scale between pixels, enabling ImageJ 1.49v software (National Institutes of Health, Bethesda, MD, USA) to calculate the wound area. Gross wound changes were documented, encompassing appearance, size, crust formation time, and detachment.

## Sample collection

The animals underwent euthanasia on specific dates (3, 6, 9, 12, 15, 18, 21, 24, 27, and 30) following the induction of burning. This euthanasia method entailed an intramuscular (IM) injection of xylazine–ketamine (91 mg/kg IM of 10% ketamine + 9.1 mg/kg IM of 2% xylazine). Upon confirmation of profound anesthesia through the absence of pedal withdrawal and corneal reflexes, cardiac puncture exsanguination was performed under sustained anesthesia to ensure the animal's demise without any possibility of recovery.

The entire thickness of the wound tissue, with a minimum of 0.5 cm of normal tissue borders, was subsequently collected for histological research. The samples were then used specifically for paraffin embedding and staining with hematoxylin–eosin (H&E), followed by subsequent histopathological evaluation. Additionally, approximately 100 mg of skin tissue (from the mid and lateral burned regions) was aseptically collected and rapidly frozen on-site with liquid nitrogen to prevent RNA degradation. These samples were used to assess the alterations in mRNA gene expression levels.

## Histopathological evaluation

The collected tissues underwent histological assessment in a randomized, single-blind format by a certified anatomic veterinary pathologist from the Veterinary Pathology Laboratories at Jordan University of Science and Technology. Samples from three rats in each group were taken and vertically embedded in paraffin blocks every three days. Light microscopy was conducted on 4- to 5-μm-thick paraffin sections stained with hematoxylin and eosin, as well as Masson's triple stain, utilizing a Nikon Eclipse 400 equipped with a Colpix 4500 digital camera attachment. The stained tissue sections underwent histological evaluation to determine collagen deposition, granulation, and inflammation, employing the scoring system outlined in Table 1.

## RNA extraction and cDNA synthesis

Total RNA was extracted from skin samples, specifically from the mid and lateral burned regions, using Direct-Zol™ RNA MiniPrep (Zymo Research, Irvine, CA, USA) in conjunction with TRI Reagent® (Zymo Research, Irvine, CA, USA), as per the manufacturer's protocol. RNA concentrations were quantified using a Biotek PowerWave XS2 Spectrophotometer (BioTek Instruments, Inc., Winooski, VT, USA). Two micrograms of total RNA from each sample were utilized in the reverse transcription reaction, employing the SuperScript IV VILO Master Mix (Invitrogen, Carlsbad, CA, USA).

## Real-time RT-PCR analysis and relative mRNA quantitation

The QuantiFast SYBR Green PCR Kit (Qiagen, Hilden, Germany) was employed to conduct real-time RT-qPCR analyses utilizing a Rotor-Gene Q MDx 5plex equipment. The 20 μL reaction mixture comprised 10 μL of the master mix, 1.2 μL of each forward and reverse primer (12 pmol), 1 μL of sample cDNA, and 7.8 μL of nuclease-free water. The PCR cycle parameters were as follows: hold at 95°C for 5 minutes, followed by 40 cycles of 95°C for 10 seconds, 55°C for 30 seconds, and 72°C for 10 seconds, and a final melting step at 95°C for 20 seconds.

Fluorescence intensities were measured during the annealing and extension phases of the primer extension cycles. Internal controls used in this study (to which gene expression was normalized) were β-actin and

**Table 1. Histopathological scoring system used to assess the healing of skin lesions.**

| Histopathological parameters | Score |
|---|---|
| **Re-epithelialization** | |
| None | 0 |
| Scant | 1 |
| Partial | 2 |
| Complete but with immature epithelium | 3 |
| Complete with mature epithelium | 4 |
| **Granulation tissue** | |
| None | 0 |
| Immature | 1 |
| Low amount (Mature) | 2 |
| Moderate degree of maturation | 3 |
| Mature | 4 |
| **Collagen accumulation** | |
| None | 0 |
| Scant | 1 |
| Low amount | 2 |
| Moderate | 3 |
| High amount | 4 |
| **Inflammatory cell** | |
| None | 0 |
| Minimal | 1 |
| Low amount | 2 |
| Moderate | 3 |
| High amount | 4 |
| **Angiogenesis** | |
| None | 0 |
| ≤5 veins | 1 |
| 6–19 veins | 2 |
| 10–15 veins | 3 |
| ≥15 | 4 |

glyceraldehyde-3-phosphate dehydrogenase (GAPDH). Melting curves were employed to validate the specificity of single-target amplification. Relative quantitation of gene expression was computed automatically. Triplicates from each cDNA library were analyzed. The melting curve confirms the specificity of single-target amplification. Relative quantitation was computed automatically. Primer sequences are presented in **Table 2**.

## Statistical analysis

All statistical analyses were conducted using IBM SPSS Statistics 27.0 (IBM, Chicago, IL, USA). The comparison of measured burned area, histopathology scores, and mRNA levels is presented as means ± standard deviation. A one-way analysis of variance (ANOVA) was employed to assess the assessed burn area and mRNA fold changes within the treatment groups (control vs. TM) at various time intervals (days 3, 6, 9, 12, 15, 18, 21, 24, 27, and 30 following the induction of burning). Additionally, a one-way ANOVA was utilized to compare histopathological scores on Days 3, 15, and 24 across three groups. Parametric differences were considered statistically significant at a p-value less than 0.05.

**Table 2. Primers sequences that are used in the real-time RT-qPCR analysis.**

| Gene | Forward Primer (5′→3′) | Reverse Primer (5′→3′) | Accession number |
|---|---|---|---|
| IGF-1 | AAAGTCAGCTCGTTCCATCC | GTTTCCTGCACTTCCTCTACTT | X06043 M17714 |
| PDGF-AA | CACACGCCAGACTGTGTATAA | CATGGTGATGCCTTTGTTTCTC | L41623 |
| PDGF-BB | GAGCCAAGACACCTCAAACT | ATCTCCTTCAGTGCCTTCTTG | PQ117545.1 |
| PDGF-DD | CCATTCGCAGGAAGAGAAGTAT | GCTCCGAGGTATCTCGTAAATG | AB003156.1 |
| TGF-β | CTGAACCAAGGAGACGGAATAC | GTTTGGGACTGATCCCATTGA | M55431.1 |
| Keratinocyte FGF | AGCGACACACGAGAAGTTATG | CCTTTCACTTTGCCTCGTTTG | X56551 |
| FGF7 | AGCGACACACGAGAAGTTATG | CCTTTCACTTTGCCTCGTTTG | NM_002009.4 |
| VEGF | CAATGATGAAGCCCTGGAGT | TCTCCTATGTGCTGGCTTTG | AY378102.1 |
| FGF2 | GACCCACACGTCAAACTACA | GCCGTCCATCTTCCTTCATAG | NM_001077585.1 |
| β-Actin | ACAGGATGCAGAAGGAGATTAC | ACAGTGAGGCCAGGATAGA | EF156276.1 |
| GAPDH | ACTCCCATTCTTCCACCTTTG | CCCTGTTGCTGTAGCCATATT | AF106860.2 |

## Results

In contrast to the rats in group 1 (negative control) and the rats in group 2 (positive control), the rats in group 3 (the tested group treated with Remanco) consistently demonstrated accelerated wound healing when compared to both control groups, as illustrated in Fig 1. In the rats of group 3, inflammation appeared to have significantly decreased between days 3 and 6, and there were no indications of pus discharge in any of the analyzed wounds. By the midpoint of the second week, specifically on day 9 for group 2, the symptoms of inflammation also appeared to have diminished, and no pus discharge was observed in any of the evaluated wounds. The wounds treated with Remanco skin repair plus ointment exhibited minimal to no dry crusts and exudation. In comparison with the treatments administered in groups 1 and 2, the dry crusts or exudates that emerged from the wounds treated with the red formulation appeared to be easily removed during the application of the treatment.

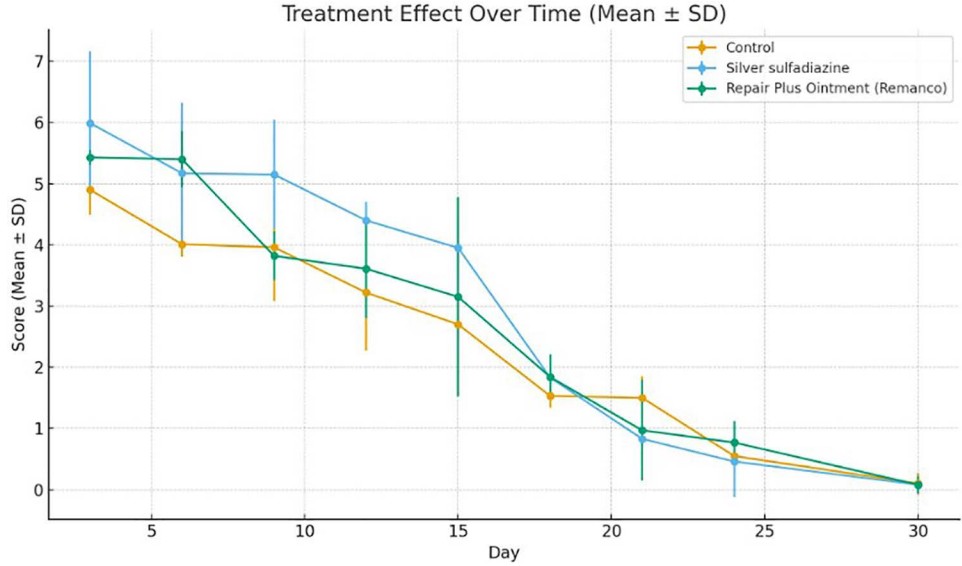

**Fig 1. Comparison of measured burned area across the three groups.** (Group 1: Normal saline; Group 2: 1% Silver sulfadiazine; Group 3: Remanco skin repair plus ointment) (n = 3 per group per time point).

Conversely, three out of the thirty rats in Group 3 displayed fully healed hairy skin by day 18 Fig 2 closely similar to the histologically normal skin Fig 3, in contrast to the skins of other groups Fig 4 A significant difference was observed among the groups concerning the duration of burn wound contraction, particularly on day 21. The results are depicted in Fig 1.

Our results indicate no significant differences among the three groups regarding the measured burned area on days 3, 6, 9, 15, 18, 21, 24, and 30 post-burn, as illustrated in Fig 1. The one-way ANOVA test results for histopathology scores also indicated no significant differences among the groups, as shown in Fig 5.

Nevertheless, the healing of the wound area exhibited a more significant improvement in the group treated with Remanco cream compared to the other groups, as illustrated in Fig 6.

The results indicate that the group administered Remanco cream exhibited a significant increase in the expression levels of two growth factors associated with the healing process: TGF-β on day 3 and IGF-1 on day 9. Nevertheless, no significant differences in gene expression were detected among the other selected genes: PDGFAA, PDGFBB, PDGFDD, Kera-FGF, VEGF, and FGF2. The results are illustrated in Fig 7A–7D and Fig 8A–8D. Although statistically significant increases in gene expression for these genes were not observed, it is evident that the fold expression of these genes

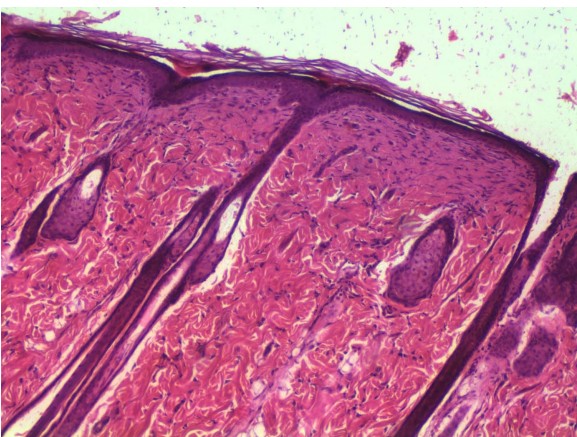

**Fig 2. Rat, skin.** The skin exhibited complete epithelialization by day 18 with relatively normal epidermal and dermal layers. H&E. 10X.

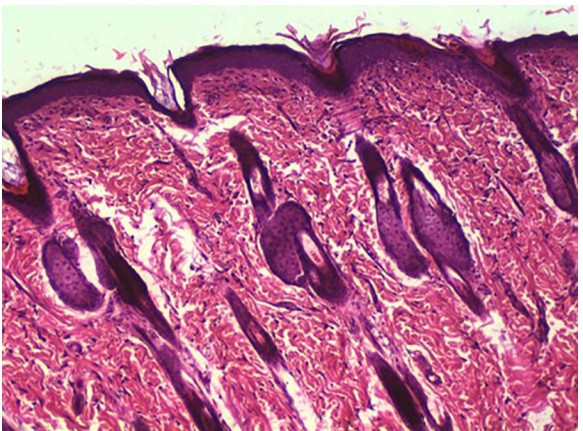

**Fig 3. Rat, skin.** Normal. Well-organized epidermal and dermal layers with regular dermal collagen structure, and intact adnexal structures (glands and follicles). No significant histopathological changes are present. H&E. 10X.

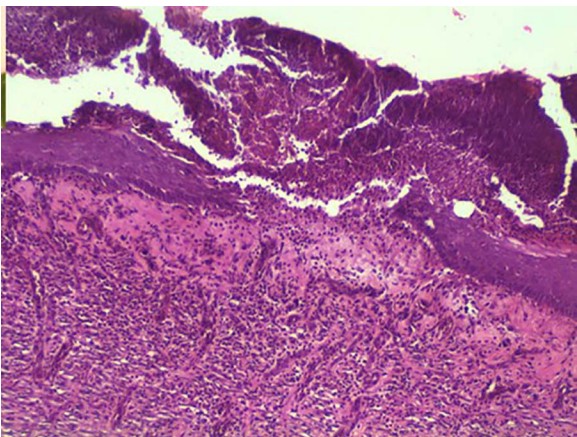

**Fig 4. Rat, skin.** The skin exhibited incomplete epithelialization by day 18 with presence of serocellular crust, granulation tissues and inflammatory cell infiltrates. H&E. 10X.

increased at various time points; specifically, the expression of TGF-β also increased two-fold on day 9 and threefold on day 18 (Fig 7A). The expression of IGF-1 was three times greater than that of the control on day 24 (Fig 7B). VEGF exhibited nearly a two-fold increase on both days 12 and 24 (Fig 7C). PDGFRa showed a two-fold increase on day 9 and a 2.3-fold increase on day 21 (Fig 7D). FGF-7 increased by approximately 1.5-fold on days 9, 15, 9,15,21, and 24 (Fig 8A). FGF-2 increased four-fold on day 21 (Fig 8C). PDGFb exhibited a 2.5-fold increase on day 3 and approximately a two-fold increase on both days 9 and 18 (Fig 7D).

Relative quantification was computed automatically based on the fluorescence signals. β-Actin and Glyceraldehyde-3-Phosphate Dehydrogenase (GAPDH) functioned as internal controls for normalizing the fold variations in gene expression. All samples were measured in triplicate. The control group was treated with normal saline, the SLV group was treated with 1% silver sulfadiazine, and the SR group was referred to as the group treated with Remanco cream.

Relative quantification was computed automatically based on the fluorescence signals. β-Actin and Glyceraldehyde-3-Phosphate Dehydrogenase (GAPDH) functioned as internal controls for normalizing the fold variations in gene expression. All samples were measured in triplicate. The control group was treated with normal saline, the SLV group was treated with 1% silver sulfadiazine, and the SR group was referred to as the group treated with Remanco cream.

## Discussion

For several decades, the traditional Jordanian alternative medicine cream utilized for burn treatment has garnered a well-deserved reputation for its efficacy in wound care. However, to our knowledge, comprehensive studies are still required to elucidate its therapeutic effects and underlying mechanisms of action. Consequently, our research objective was to assess the impact of Remanco Skin Repair Plus ointment on the morphological alterations during the wound healing process. Topical wound treatment should encompass a diverse range of antibacterial properties, in addition to its ability to shield the wound site from dehydration and other external factors, thereby expediting the healing trajectory. Conversely, the majority of synthetic chemical agents commonly employed for the treatment of open wounds may also impede the healing process [18–20]. Consequently, it is hypothesized that alternative traditional natural products may provide protection against infection in burned wounds and enhance the healing process without inducing the adverse effects associated with synthetic chemicals.

Our research demonstrated that Remanco skin repair plus ointment (group 3) exhibited a statistically significant acceleration in wound healing in rats when compared to both the negative control (group 1) and the positive control (group 2).

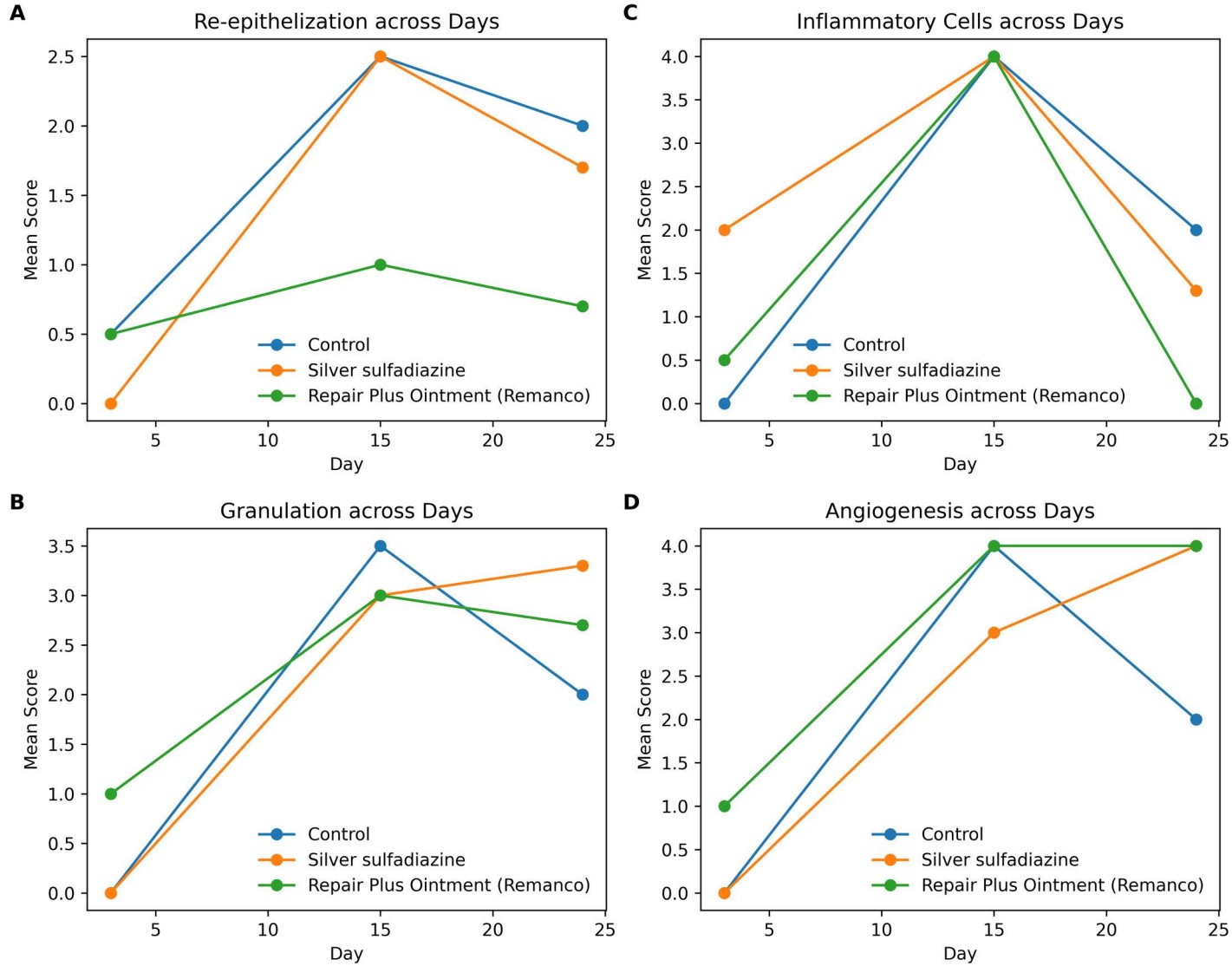

**Fig 5. Comparison of histology scores across the three groups on Days 3, 15, and 24.** (Group 1: Normal saline; Group 2: 1% Silver sulfadiazine; Group 3: Remanco skin repair plus ointment) (n = 3 per group per time point). * One-way ANOVA, a p-value for F (2, 4) for Day 3, F (2, 3) for Day 15, and F (2, 6) for Day 24.

Inflammation decreased between days 3 and 6 in group 3 and around day 9 in group 2. Inflammation is a crucial phase of wound healing, during which immune cells are recruited to the wound site to prevent infection and prepare for tissue regeneration. The faster reduction in inflammation in group 3 indicates that Remanco may include anti-inflammatory compounds. Early resolution of inflammation can hasten wound healing by reducing tissue damage and facilitating the subsequent stages of healing.

Wounds in group 3 had fewer dry crusts and exudation than those in groups 1 and 2. Maintaining an ideal moisture balance in the wound bed is critical for effective healing. Moist surroundings encourage epithelial cell migration and faster wound closure, whereas dry crusts might inhibit healing. The lower prevalence of dry crusts and exudation in group 3 shows that Remanco promotes a more favorable moisture environment, possibly by including hydrating agents or occlusive qualities that inhibit desiccation.

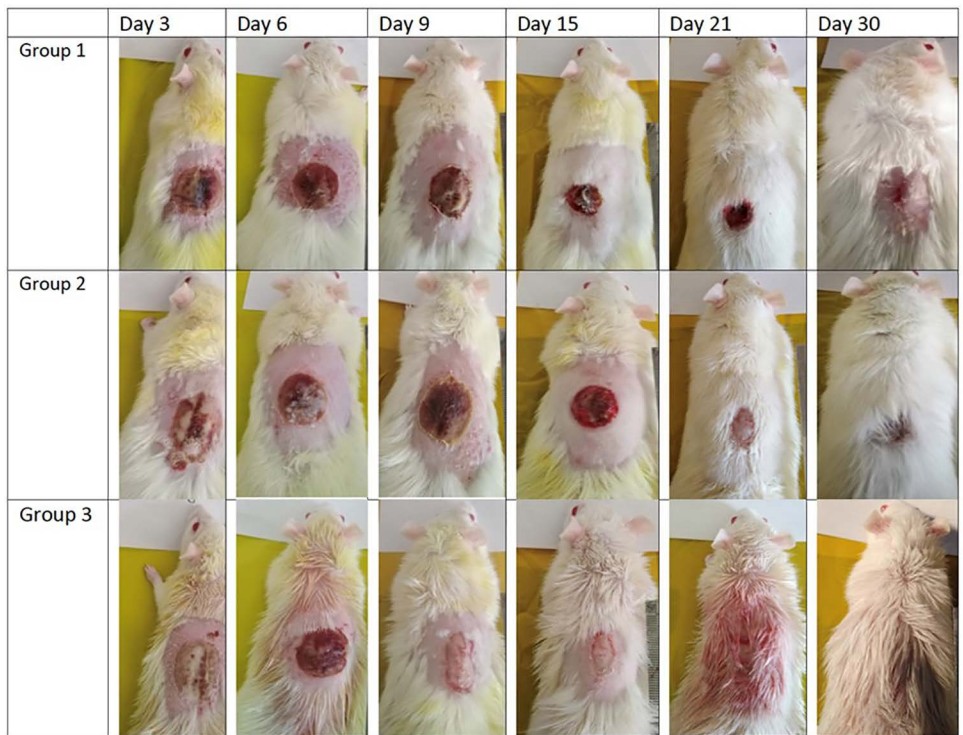

**Fig 6. Photograph of wound healing progression in three experimental groups over 30 days.** Representative images show the gross appearance of wounds on Days 3, 6, 9, 15, 21, and 30 post-treatment in Group 1, Group 2, and Group 3 rats.

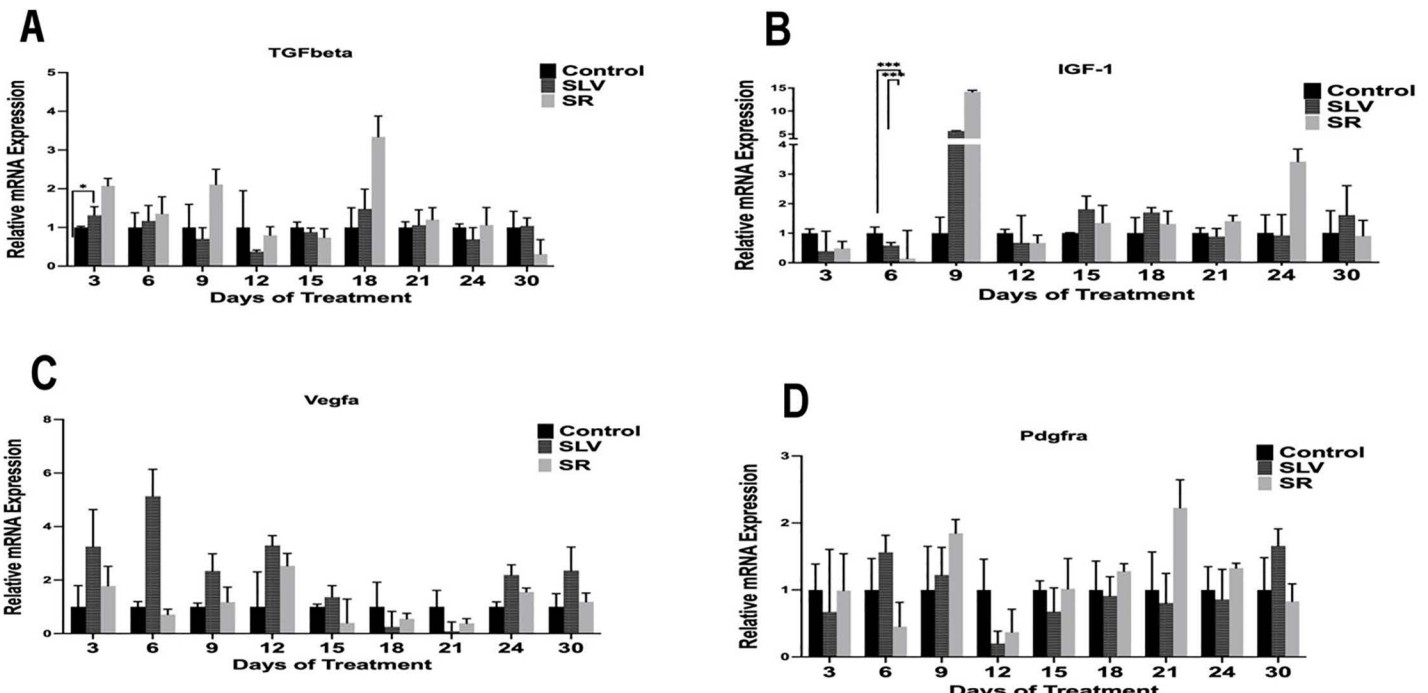

**Fig 7. (A-D): Levels of fold expression of TGF-beta (A), IGF-1 (B), VEGFA (C), PDGFra (D).** (n = 3 per group per time point).

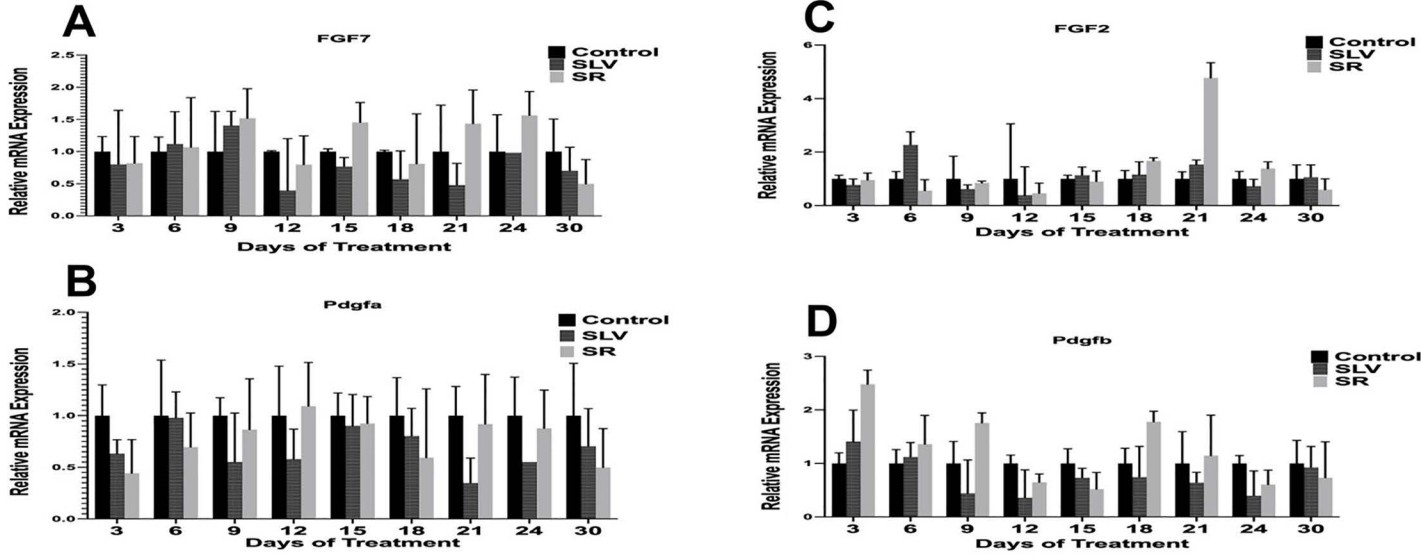

**Fig 8. (A-D): Levels of fold expression of FGF7 (A), FGF2 (B), PDGFa (C), and PDGFb (D).** (n = 3 per group per time point).

By day 18, three of the thirty rats in group three had entirely healed, with hairy skin, which was not seen in the other groups. The presence of hairy skin on day 18 suggests complete re-epithelialization and the start of tissue remodeling (Figs 2–4). This rapid healing shows that Remanco not only accelerates wound closure but also stimulates hair follicle regeneration. In various wound therapies, growth factors and bioactive substances can promote cell proliferation and differentiation, resulting in faster wound closure and tissue regeneration. This could be because some components in Remanco boost these restorative processes.

The lack of statistically significant changes could be attributed to the sensitivity of the measurement technique or the study's sample size. Slight differences in wound healing may be statistically insignificant if the sample size is insufficient or the assessment methods are insensitive enough to identify minor changes. Previous research has emphasized the need to utilize sensitive and reproducible wound assessment methods to detect modest but clinically significant variations [21]. Increasing the sample size or employing more advanced imaging techniques may show substantial differences that were not identified in the current investigation.

Our findings indicate that treatment with Remanco cream significantly enhances the expression levels of two crucial growth factors integral to the healing process: TGF-beta on day 3 and IGF-1 on day 9. These data demonstrate a substantial early response to Remanco cream, which may be pivotal in expediting the initial stages of wound healing. TGF-beta, widely recognized for its role in regulating inflammation and facilitating tissue regeneration, exhibited a notable increase in expression levels during the early stages of the therapy course. On days 9 and 18, TGF-beta expression doubled and tripled, respectively (Fig 7A). This increase is significant because TGF-beta is essential for orchestrating the deposition of extracellular matrix and the formation of granulation tissue, both of which are critical phases in the healing process [22].

Similarly, IGF-1, a growth factor involved in cellular proliferation and differentiation, increased significantly on day 9, with a threefold rise compared to the controls on day 24 (Fig 7B). This prolonged increase in IGF-1 expression indicates that Remanco cream triggers a fast-healing response and promotes long-term tissue repair and regeneration [23].

Although the expression levels of other selected genes, such as PDGFAA, PDGFBB, PDGFDD, Kera-FGF, VEGF, and FGF2, were not statistically significant, clear trends were observed, indicating higher fold expression at various time

intervals. For instance, on days 12 and 24, VEGF, which is essential for angiogenesis, increased nearly twofold (Fig 7C). Although not statistically significant, this trend highlights Remanco cream's potential to enhance vascularization in the wound area, which is crucial for efficient wound healing [24,25]. Similarly, PDGFRa increased two-fold on day 9 and 2.3-fold on day 21 (Fig 7D), indicating a function in increasing the recruitment and proliferation of cells required for wound healing. FGF-7, shown to stimulate epithelial cell proliferation, increased 1.5-fold on days 9, 15, 21, and 24 (Fig 8A), indicating possible benefits for wound re-epithelialization [26,27]. Furthermore, on day 21, FGF-2, which is involved in angiogenesis and tissue repair, increased fourfold (Fig 8C), highlighting the significance of Remanco cream in boosting several elements of the healing process.

PDGFb, another crucial growth factor, exhibited a 2.5-fold increase on day 3 and nearly doubled on days 9 and 18 (Fig 8D). This expression pattern suggests a robust early and sustained response, potentially enhancing the recruitment and activation of fibroblasts and other cells essential for wound healing [28,29].

So, in summary, while the study highlights those big jumps in TGF-beta and IGF-1, the way other growth factors are being produced suggests that Remanco cream might be doing more than just that for wound healing. To truly understand what's happening at the molecular level and make Remanco cream even more effective, we need to conduct further research.

## Supporting information

**S1 File. Supporting data.** The datasets supporting this study have been deposited in Figshare and are publicly available at 10.6084/m9.figshare.30598193.
(ZIP)

## Author contributions

**Conceptualization:** Foad Alzoughool, Reman Alnemrat, Mohammad Alzghool.

**Data curation:** Foad Alzoughool, Manar Atoum, Mohammad Alzghool.

**Formal analysis:** Ahmad M. Al-Bashaireh.

**Funding acquisition:** Foad Alzoughool.

**Investigation:** Foad Alzoughool, Mohammad Borhan Al-Zghoul, Wael Hananeh, Mohamad Mayyas.

**Methodology:** Foad Alzoughool, Mohammad Borhan Al-Zghoul, Wael Hananeh, Mohamad Mayyas.

**Resources:** Manar Atoum, Reman Alnemrat.

**Software:** Mohammad Borhan Al-Zghoul.

**Supervision:** Foad Alzoughool.

**Validation:** Foad Alzoughool, Manar Atoum, Yousef Aljawarneh.

**Visualization:** Yousef Aljawarneh.

**Writing – original draft:** Foad Alzoughool.

**Writing – review & editing:** Mohammad Borhan Al-Zghoul, Mohammad Alzghool, Ahmad M. Al-Bashaireh, Yousef Aljawarneh.

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
