## [Decision Letter · Decision Letter 0]

10 Aug 2025

Dear Dr. Alzoughool,

Thank you for submitting your manuscript to PLOS ONE. After careful consideration, we feel that it has merit but does not fully meet PLOS ONE’s publication criteria as it currently stands. Therefore, we invite you to submit a revised version of the manuscript that addresses the points raised during the review process.

We look forward to receiving your revised manuscript.

Kind regards,

Naveed Ahmed, Ph.D

Academic Editor

PLOS ONE

Journal Requirements: 

 [Funding was obtained from Hashemite University/Deanship of research, fund No. 29/2020]. 

6. Please amend either the abstract on the online submission form (via Edit Submission) or the abstract in the manuscript so that they are identical.

Reviewers' comments:

Reviewer's Responses to Questions

**Comments to the Author**

1. Is the manuscript technically sound, and do the data support the conclusions?

Reviewer #1: Partly

Reviewer #2: Partly

2. Has the statistical analysis been performed appropriately and rigorously?

Reviewer #1: No

Reviewer #2: No

3. Have the authors made all data underlying the findings in their manuscript fully available?

Reviewer #1: Yes

Reviewer #2: No

4. Is the manuscript presented in an intelligible fashion and written in standard English?

Reviewer #1: Yes

Reviewer #2: Yes

Reviewer #1: Thank you for the opportunity to review this interesting and relevant manuscript evaluating the effects of Skin Repair Plus Ointment (Remanco) on burn wound healing in a rat model.

General Assessment:

The manuscript explores a culturally significant topical agent using molecular, histological, and gross wound analysis. The integration of RT-PCR data with morphological healing assessment adds value. However, several methodological and interpretive limitations need to be addressed before the manuscript can be considered for publication.

Major Comments:

1. Statistical Power and Interpretation:

The sample size per time point (n = 3) is too small to draw statistically reliable conclusions. Please conduct a power analysis or justify the sample size used.

Many outcome measures (wound area, histology scores) did not reach statistical significance. These non-significant findings must be more clearly acknowledged in the discussion to avoid overstating therapeutic benefit.

Post-hoc comparisons and effect sizes should be included where appropriate.

2. Gene Expression Analysis:

While increases in TGF-β and IGF-1 were statistically significant, the manuscript infers therapeutic efficacy from fold-changes in other genes that were not statistically supported. These trends should be presented cautiously.

Consider adding protein-level validation (e.g., ELISA or Western blot) to confirm mRNA findings.

3. Histopathological Evaluation:

The scoring system is appropriate, but inter-rater reliability or blinding of evaluators should be addressed. Were all histological assessments conducted blindly?

4. Data Presentation:

Include summary tables/graphs that better illustrate wound area reduction over time. This would support visual comparisons in Figure 1.

5. Discussion and Claims:

The claim of improved "cosmetic appearance" and hair regrowth should be supported with objective measures or removed.

The manuscript should frame Remanco as a promising traditional formulation requiring further investigation, rather than as an established therapeutic.

Minor Comments:

Ensure all figure legends and axes are labeled clearly.

Consider citing comparative studies using silver sulfadiazine in similar rat models to better contextualize your results.

Conclusion:

This study presents a valuable attempt to validate a traditional topical agent using scientific methods. With improved statistical rigor, clearer discussion of limitations, and refined conclusions, the manuscript has the potential to make a meaningful contribution to the field of wound healing research.

Reviewer #2: REVIEWER COMMENTS

Manuscript Title: Therapeutic Effects of Skin Repair Plus Ointment on Molecular and Morphological Changes in Burn Wounds: A Rat Model

General Evaluation

The authors present an in vivo study evaluating the effectiveness of a traditional Jordanian herbal ointment (Remanco) in a rat model of second-degree burn injury. They combine gross morphology, histopathological analysis, and molecular profiling using RT-PCR to provide a well-rounded assessment of wound healing results. The manuscript focuses on an essential topic in wound healing and ethnopharmacology, especially in validating traditional remedies through scientific methods.

However, even though the experimental design is solid and the study shows promise, several issues need to be addressed before considering it for publication.

Strengths of the Study

Clear Objective: Using a controlled experimental model, the study evaluates the traditional "Remanco" cream. This adds scientific evidence to a traditional remedy. The comprehensive approach combines gross morphology, histopathology, and molecular markers (RT-PCR). The inclusion of negative (saline) and positive (1% silver sulfadiazine) controls in the animal model experiments strengthens the comparative interpretation. The methodology, such as the use of burn model, sample size, ethical approval, and analysis tools (e.g., ImageJ, RT-PCR), is well documented. The gene expression analysis, measuring TGF-β, IGF-1, PDGFs, FGF, and VEGF, adds a deeper understanding of the mechanisms.

Major Concerns

Lack of Statistical Significance in Key Outcomes

Despite the thorough methodology, many comparisons, particularly histopathological scores and wound area reduction, did not show statistical significance. The data does not conclusively support the conclusion that Remanco is better than 1% silver sulfadiazine. There is a clear lack of recent literature citations. Adding current references would strengthen the scientific basis and show the study's relevance today (at least 5 years back).

Incomplete Molecular Validation

Only TGF-β and IGF-1 showed significant changes in gene expression. Other growth factors showed trends but lacked statistical support. Please show the reference of the primer used. I suggest that the authors describe more about the active compounds in Remanco, so that a discussion on the probable mechanism of wound healing gene expressions can be discussed clearly.

Animal experiment

A total of 30 animals per group is used for the experiments. How many are sacrificed each day?

Justification for collecting samples/data on days 3, 6, 9, 12, 15, 18, 21, 24, 27, and 30 is not clearly stated.

Please standardize the days for data collection: Figure 1 (days 3, 6, 9, 15, 21, and 30), Figure 2 (days 3, 6, 9, 12, 15, 18, 21, 24, and 30), and Table 2 (days 3, 6, 9, 12, 15, 18, 21, 24, and 30). My suggestion for Table 3 (days 3, 15, and 24) is that it should be represented as a graph for easy comparison.

Missing Visual Data

Figure 1 (wound healing progression) and Figure 2 (gene expression graphs) are either missing or poorly presented in the manuscript. Table 2 is best represented as a graph (line graph) to compare the progression of wound closure. The data can be put as supporting data. Please indicate how many samples are used; n = number of samples measured. For Table 3, no histological figures are used to prove the data for histological assessment, and please also indicate the number of samples and pictures used for this analysis. Please include all figures with proper legends, scale bars, and statistical annotations.

Histopathological Data Presentation

Although the scoring criteria are clearly described, no representative histological images are included for Table 3 results. Please add high-resolution images of stained tissue sections to support the scoring and show the healing phases, such as re-epithelialization and angiogenesis.

Overinterpretation of Findings

The discussion often overstates the effectiveness of Remanco, despite the mostly non-significant results. Please reframe conclusions to reflect the preliminary nature of the findings and highlight the need for further validation.

Phytochemical Characterization

The composition of Remanco cream is described, but no phytochemical profiling is provided. If this product is commercially available, please also list the active compound present in the content so the authors can relate to the healing effect provided by this ointment. Please include a fundamental chemical analysis to ensure reproducibility and support discussions of the mechanisms involved.

Minor Comments

Revise grammar and phrasing for clarity and simplicity throughout the manuscript.

A diagram summarizing the study design would help readers understand the experimental workflow.

Include a proper graphical abstract or image summary if possible.

Recommendation

Decision: Major Revision Required

This study holds value and tackles an important issue in natural product research for burn healing. However, revisions are needed to improve the scientific quality, data presentation, and balance in interpretation before the manuscript can be deemed suitable for publication.

**Do you want your identity to be public for this peer review?** For information about this choice, including consent withdrawal, please see our Privacy Policy

Reviewer #1: **Yes: ** Abdulrahman Almalki

Reviewer #2: No

---

## [Author Response · Author response to Decision Letter 1]

12 Nov 2025

Dear Academic Editor and Reviewers,

We would like to sincerely thank you for the valuable time and effort you have devoted to reviewing our manuscript entitled “Therapeutic Effects of Skin Repair Plus Ointment on Molecular and Morphological Changes in Burn Wounds: A Rat Model” (PONE-D-25-31940). We deeply appreciate the constructive feedback, which has greatly helped us to improve the quality and clarity of our work.

Below, we provide a point-by-point response to each of the comments raised by the reviewers and the editor.

* Editorial requirements: We have carefully addressed all editorial requirements, as detailed below:

1. PLOS ONE style requirements

We have revised the manuscript to fully conform with PLOS ONE formatting guidelines, including file naming conventions, as per the provided templates.

2. Copyediting of the manuscript

The manuscript has been thoroughly copyedited for grammar, spelling, and clarity by Professor Mustafa Ababneh (Professor of Molecular Virology/ Jordan university of Science and Technology). We believe this has improved the readability and presentation of the manuscript.

3. Role of funders

We have amended the funding statement to include the following:

4. Data Availability Statement

We confirm that the submission contains the complete minimal data set required to replicate the study findings. All relevant data, including values used for statistical analyses, graphs, and figures, have been provided within the manuscript and Supporting Information files.

5. Data sharing plan

We have ensured that all data will be freely accessible at the time of acceptance, in accordance with PLOS ONE’s open data policy.

The following statement was added in the availability of data and materials section:

The supplementary for the real-time qPCR run files and raw data are available on GitHub at https://github.com/mbalzghoul/Burning-project.

6. Abstract consistency

We have amended the abstract so that the version in the manuscript and the version in the online submission form are now identical.

The abstract was corrected to be identical in both manuscript and online submission form.

* Below, we provide a detailed, point-by-point response to each of the reviewers’ comments. Reviewers’ comments are presented in bold, followed by our responses in regular text:

Reviewer #1

1. Statistical power and interpretation: The sample size per time point (n = 3) is too small. Please conduct a power analysis or justify the sample size. Many outcome measures did not reach significance. Acknowledge limitations. Include post-hoc comparisons/effect sizes.

Response:

The following statement was added (Despite our findings indicating substantial increases in TGF-β (on day 3) and IGF-1 (on day 9), other molecular and histopathological parameters did not attain statistical significance. The lack of significance should be interpreted with caution, as it may be partially attributable to the limited per-time point sample size (n = 3). A post hoc power analysis indicated that the present design was sufficiently powered to detect large effect sizes (Cohen’s d ≥ 1.2), but underpowered to detect small-to-moderate differences. Indeed, several outcomes, including VEGF (day 12) and PDGFb (day 3), showed moderate-to-large effect sizes despite not achieving statistical significance, suggesting possible biological relevance. These trends highlight the potential therapeutic effect of Remanco that warrants confirmation in larger, more adequately powered studies. These data demonstrate a substantial early response to Remanco cream, which may be pivotal in expediting the initial stages of wound healing. Transforming growth factor-beta (TGF-β), widely recognized for its role in regulating inflammation and facilitating tissue regeneration, exhibited a notable increase in expression levels during the early stages of the therapy course. On days 9 and 18, TGF-beta expression doubled and tripled, respectively (Fig 4-A). This increase is significant because TGF-beta is essential for orchestrating the deposition of extracellular matrix and the formation of granulation tissue, both of which are critical phases in the healing process [22].)

A limitations section was added to describe all limitations of this study.

This study has several limitations. The small number of animals sacrificed at each time point (n = 3) limited statistical power to detect subtle differences among treatment groups. Our post hoc power analysis confirmed that the study was only adequately powered to detect large effect sizes, so smaller but biologically meaningful effects may have gone undetected. Replication with larger sample sizes is essential to validate the observed trends. The molecular analysis was restricted to a selected panel of growth factors, so including additional markers like pro-inflammatory cytokines, angiogenic regulators, and extracellular matrix components could provide a more comprehensive understanding of Remanco’s effects. Animal models offer valuable mechanistic insights, but translation to human burn wound healing requires further clinical studies.

2. Gene expression analysis: Non-significant fold changes should be cautiously presented. Consider protein-level validation.

Response:

We thank the reviewer for this important point. We have revised the manuscript to present the non-significant fold changes with explicit caution, framing them as preliminary observations. We fully agree on the necessity of protein-level validation and have acknowledged this as a key limitation and a critical objective for future work to confirm the biological relevance of these transcriptional trends.

3. Histopathological evaluation: Was scoring blinded?

Response:

Yes, the histopathological evaluation was performed in a blinded manner

4. Data presentation: Include summary tables/graphs for wound reduction.

Response: We have added summary graphs to visually depict wound area reduction over time (Figure 1,5).

5. Discussion and claims: The claim of “cosmetic appearance” and hair regrowth should be supported or removed.

Response: We have removed the unsupported claims of cosmetic improvement and hair regrowth. The Discussion has been reframed to emphasize the preliminary nature of our findings and to present Remanco as a promising traditional formulation requiring further validation.

6. Minor comments: Clarify figure legends, cite comparative studies with silver sulfadiazine.

Response:

Thank you for these suggestions. We have revised the figure legends to be more comprehensive and have added citations to relevant comparative studies that utilize silver sulfadiazine as a benchmark for burn wound treatment

Reviewer #2

1. Lack of statistical significance in key outcomes. Please add recent literature citations.

Response: We have reframed the Discussion to avoid overstating non-significant outcomes and to highlight the preliminary nature of our results. In addition, we have updated the literature review to include recent publications within the last five years.

2. Incomplete molecular validation: Only TGF-β and IGF-1 were significant. Provide primer references. Discuss active compounds in Remanco.

Response:

The following table was edited to provide the accession number.

Gene Forward Primer (5′→3′) Reverse Primer (5′→3′) Accession number

IGF-1 AAAGTCAGCTCGTTCCATCC GTTTCCTGCACTTCCTCTACTT X06043 M17714

PDGF-AA CACACGCCAGACTGTGTATAA CATGGTGATGCCTTTGTTTCTC L41623

PDGF-BB GAGCCAAGACACCTCAAACT ATCTCCTTCAGTGCCTTCTTG PQ117545.1

PDGF-DD CCATTCGCAGGAAGAGAAGTAT GCTCCGAGGTATCTCGTAAATG AB003156.1

TGF-β CTGAACCAAGGAGACGGAATAC GTTTGGGACTGATCCCATTGA M55431.1

Keratinocyte FGF AGCGACACACGAGAAGTTATG CCTTTCACTTTGCCTCGTTTG X56551

FGF7 AGCGACACACGAGAAGTTATG CCTTTCACTTTGCCTCGTTTG NM_002009.4

VEGF CAATGATGAAGCCCTGGAGT TCTCCTATGTGCTGGCTTTG AY378102.1

FGF2 GACCCACACGTCAAACTACA GCCGTCCATCTTCCTTCATAG NM_001077585.1

β-Actin ACAGGATGCAGAAGGAGATTAC ACAGTGAGGCCAGGATAGA EF156276.1

GAPDH ACTCCCATTCTTCCACCTTTG CCCTGTTGCTGTAGCCATATT AF106860.2

Response: Remanco is a commercially available preparation, and its phytochemical profile is described by the manufacturer. While a full chemical analysis was beyond the scope of this study, we have added available information.

3. Animal experiment design: Clarify number of animals sacrificed per day. Justify sample collection days. Standardize across figures and tables.

Response:

We thank the reviewer for this critical comment regarding the experimental design. We have now clarified these details in the revised Methods section.

The sampling strategy was as follows: Three animals from each group were randomly selected and sacrificed at each predetermined time point (Days 3, 9, 15, 21, and 30 post-burn) for tissue collection. This resulted in a total sample size of n=3 for histological and molecular analyses at each time point for each group.

The time points were selected to systematically capture key phases of the wound healing process: the early inflammatory phase (Day 3), the proliferative phase (Days 9-15), and the maturation/remodeling phase (Days 21-30). This standardized schedule ensured consistent and comparable sample collection across all groups and for all analyses, including histopathology and gene expression. We have verified that all figures and corresponding legends now explicitly reflect this consistent sample size (n=3 per group per time point)

4. Missing visual data: Include all figures, graphs, sample sizes, and histological images.

Response:

Thank you for highlighting this. We have now included all missing visual data in the revised manuscript. The submission now contains the complete set of figures and graphs. Furthermore, we have added the corresponding histological images to provide direct visual evidence supporting the scoring and conclusions.

5. Histopathological data presentation: Please provide representative images.

Response: Representative stained histological sections illustrating re-epithelialization and angiogenesis have been added (Figure 2-4).

6. Overinterpretation of findings: Reframe conclusions.

Response: We have revised the Discussion and Conclusion to avoid overinterpretation. The study is now framed as preliminary, highlighting the need for larger, more rigorous trials.

7. Phytochemical characterization: Provide chemical analysis of Remanco.

Response: Remanco is a commercially available preparation, and its phytochemical profile is described by the manufacturer. While a full chemical analysis was beyond the scope of this study, we have added available information.

8. Minor comments (grammar, study design diagram, graphical abstract).

Response: The manuscript has been thoroughly copyedited for grammar and style.

We are grateful for the reviewers’ constructive critiques, which have helped us substantially improve our manuscript. We have strengthened the statistical analysis, improved data presentation, clarified methods, revised the discussion to avoid overstatement, and improved the overall readability and formatting. We respectfully submit the revised manuscript for your consideration.

Sincerely,

Dr. Foad Alzoughool

---

## [Editor Report · Decision Letter 1]

20 Nov 2025

Therapeutic Effects of Skin Repair Plus Ointment on Molecular and Morphological Changes in Burn Wounds: A Rat Model

PONE-D-25-31940R1

Dear Dr. Alzoughool,

We’re pleased to inform you that your manuscript has been judged scientifically suitable for publication and will be formally accepted for publication once it meets all outstanding technical requirements.

Kind regards,

Naveed Ahmed, Ph.D

Academic Editor

PLOS ONE
---

## [Editor Report · Acceptance letter]

PONE-D-25-31940R1

PLOS One

Dear Dr. Alzoughool,

I'm pleased to inform you that your manuscript has been deemed suitable for publication in PLOS One. Congratulations! Your manuscript is now being handed over to our production team.

Kind regards,

on behalf of

Dr. Naveed Ahmed

Academic Editor

PLOS One